# Assessment of Negative Gingival Recession: A Critical Component of Periodontal Diagnosis

I-Ching Wang [1,*,†] , Hsun-Liang Chan [2], Georgia K. Johnson [1] and Satheesh Elangovan [1]

1   Department of Periodontics, College of Dentistry, University of Iowa, Iowa City, IA 52242, USA;
    georgia-johnson@uiowa.edu (G.K.J.); satheesh-elangovan@uiowa.edu (S.E.)
2   Department of Periodontics and Oral Medicine, School of Dentistry, University of Michigan,
    Ann Arbor, MI 48109, USA; hlchan@umich.edu
*   Correspondence: i-ching-wang@uiowa.edu
†   Current address: Dental Science Building, College of Dentistry, University of Iowa, 801 Newton Rd.,
    Iowa City, IA 52242, USA.

**Abstract:** Accurate measurement of negative gingival recession (GR) is essential to accurately determine the clinical attachment loss, which leads to an accurate diagnosis and optimal therapy of periodontal disease. However, the accuracy of measuring the negative GR has been shown to be low and highly variable between examiners. The position of the gingiva margin in relation to the cemento-enamel junction (CEJ) varies among different stages of passive eruption. The amount of negative GR is about 2 mm on average at the mid-facial sites and ranges from 2 to 3.5 mm at interproximal sites in periodontally healthy patients. Some other clinical conditions may change the gingival dimension coronal to the CEJ, such as altered passive eruption and gingival enlargement. In addition to the traditional approach using a periodontal probe to assess the negative GR, nowadays dental ultrasound imaging may be able to assist in accurately measuring the amount of negative GR. This narrative review will discuss the existing evidence of the dimension of dentogingival tissue and the clinical assessment of negative GR using different clinical tools.

**Keywords:** negative gingival recession; clinical attachment loss; ultrasonography

## 1. Introduction

The term, negative gingival recession (GR), indicates that the free gingival margin is coronal to the cemento-enamel junction (CEJ). It is an important component of the periodontal examination that allows the clinician to calculate the clinical attachment loss (CAL) to define disease severity and monitor disease progression. CAL represents the extent of periodontal tissue loss, which is measured as the distance from the CEJ to the base of the pocket [1] and is clinically calculated by deducting the distance of the gingival margin coronal to the CEJ from the pocket depth [2]. However, it involves some "guess work" to determine the amount of negative GR in relation to the CEJ position, and limited reproducibility and measurement errors have been reported in the literature [3–5]. Therefore, the early diagnosis of initial periodontitis can be challenging to clinicians, and the assignment of periodontitis severity and progressive changes of attachment level over time could be inconsistent. In view of these considerations, the aim of this review is to discuss the normal anatomy and pathologic alteration of the dentogingival tissues, the clinical assessment of the negative GR, and the potential aid of modern technology to accurately measure negative GR.

## 2. Normal Anatomy

Preservation of an intact dentogingival unit with the gingival margin slightly coronal to the CEJ in a state of optimal health is a hallmark feature for an intact, healthy dentition. The physiologic dimensions of a healthy periodontium can be divided into (1) superficial,

(2) crevicular, and (3) subcrevicular physiologic dimensions [6], which correspond to the oral, sulcular, and junctional epithelium with the underlying connective tissue, respectively. The biological interface between the gingiva and the tooth that forms the initial barrier to underlying tissues is known as the "dentogingival junction" (DGJ) [7]. The average dimension of sulcus depth, epithelial attachment, and the connective tissue attachment was reported to be 0.69 mm, 0.97 mm, and 1.07 mm, respectively [8]. Today, the latter two components forming a functional unit, is termed the "supracrestal tissue attachment" [9] which was previously known as the "biologic width".

The apical migration of the DGJ after dentition completes the active phase of eruption is called passive eruption [10]. Gargiulo et al. described the changes that occur in the location of DGJ in relation to the CEJ in four stages of passive eruption [8]. In stage I of the passive eruption, which represents a physiologically healthy state and usually occurs before 30 years old of age, the location of epithelial attachment (today called junctional epithelium) is entirely on the enamel with the most apical termination at the CEJ. The average gingival dimension coronal to the CEJ (negative GR) at stage I that comprises of sulcus depth and epithelial attachment is 2.15 mm and ranges from 0.28 to 6.34 mm [8]. Along with the apical shift of the dentogingival junction, which is considered a consequence of pathological periodontal destruction, the epithelial attachment is on the enamel and cementum at stage II and entirely located on the cementum at stage III until the epithelial attachment and gingival margin lie apical to the CEJ at stage IV [8,11,12]. The length of the epithelial attachment decreases as one progresses from stage I to stage IV of passive eruption, as compared to a relatively constant sulcus depth and connective tissue attachment. The average gingival dimension coronal to the CEJ (negative GR) is 1.29 mm (sulcus and part of epithelial attachment) and 0.6 mm (purely sulcus depth) at stages II and III, respectively. In stage IV and beyond, the gingival margin is at the level of or apical to the CEJ (Figure 1, Table 1). The dimension described by Gargiulo et al. (stage I–III of passive eruption) is in accordance with the common understanding that the facial gingiva margin is approximately 0.5 to 2 mm coronal to the CEJ. This was based on the observation of the distance from free gingiva margin to the free gingival groove and the latter corresponds to the bottom of the gingiva crevice (the base of the epithelial attachment) that is often located at CEJ [13].

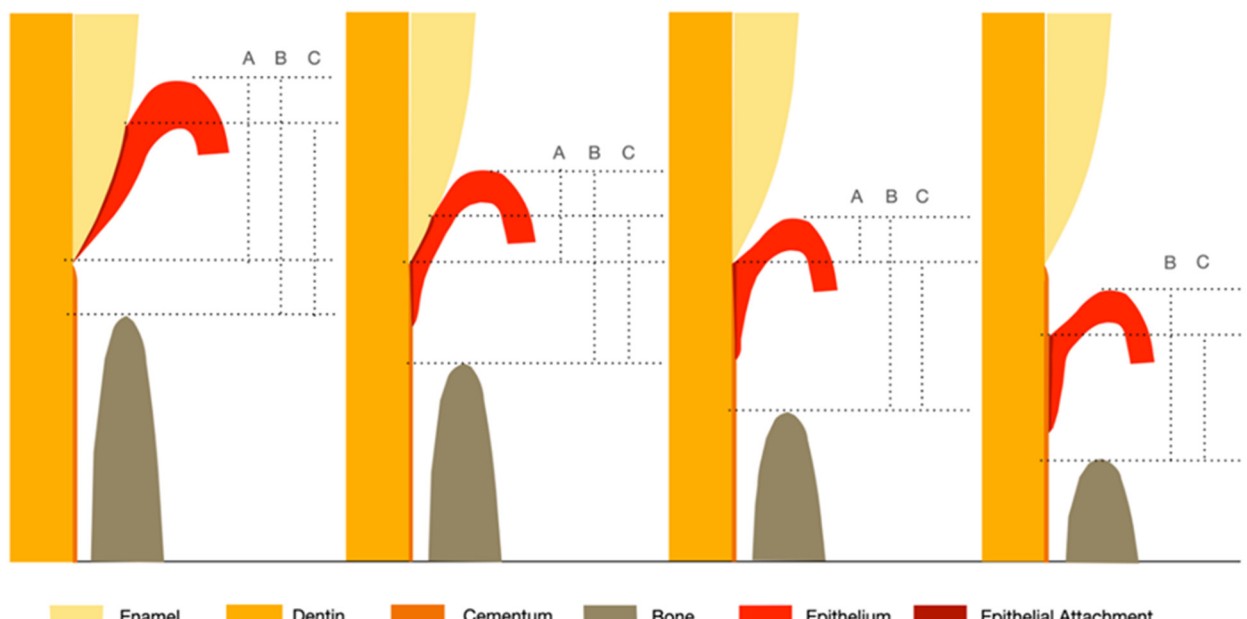

**Figure 1.** Four different stages of passive eruption. A: Negative gingival recession; B: total length of dentogingival junction, including the sulcus depth, epithelial attachment, and connective tissue attachment; C: supracrestal tissue attachment, including epithelial attachment and connective tissue attachment (adapted from Gargiulo et al., 1961 [8]).

**Table 1.** Dimensions of dentogingival junction in 4 different stages of passive eruption.

| Passive Eruption Phase | I | II | III | IV |
|---|---|---|---|---|
| Mean age in years | 24.5 | 31.4 | 32.3 | 39.7 |
| Negative gingival recession (mm) | 2.15 | 1.29 | 0.6 | - |
| Total dentogingival junction (mm) | 3.23 | 2.78 | 2.41 | 2.53 |
| Supracrestal tissue attachment (mm) | 2.43 | 2.17 | 1.8 | 1.77 |

Adapted from Gargiulo et al., 1961 [8], all values are average measurements. Note that the age of study population ranged from 19 to 50 years old.

Another relevant finding from Gargiulo's study is that the sulcus depth and epithelial attachment length are slightly greater at the interproximal sites as compared to the buccal or lingual sites, especially at stages I and II, which was confirmed in later studies [14,15]. Normally, the scalloped osseous crest parallels the CEJ circumferentially. The osseous scallop (defined by the distance from the crestal bone level at the mid-buccal/lingual site to the crestal bone level at the interproximal site) is greatest at the maxillary anterior teeth, averaging 3.5 mm, and gradually flattens out posteriorly [16]. The extent of the osseous scallop is strongly associated with the bone morphotype and can range from 2.1 mm in a flat type to as high as 4.1 mm in a pronounced scalloped type [17]. On average, the distance from the gingival margin to the bone crest is about 3 mm at the mid-facial sites and ranges from 3 to 4.5 mm at interproximal sites in periodontally healthy patients depending on the amount of gingival scallop in relation to the underlying interproximal osseous scallop [18].

The dimension of the DGJ is also associated with the gingival phenotype in that the distance of the gingiva margin to the bone crest is reported to be greater in patients with a thick (flat) phenotype than in those with a thin (pronounced scalloped) phenotype at the buccal (3.1 mm versus 2.5 mm) and interproximal sites (4.3 mm versus 3.2 mm) [19,20]. These observations are in line with the finding that papilla fill can be 100% found when the distance from contact point to bone crest is less than 5 mm [21,22]. In addition, Kim et al. reported that in patients with no interproximal bone loss that the distance from the contact point to the interproximal CEJ between maxillary central incisors was $3.73 \pm 0.98$ mm [23]. To sum up, the dimension of the DGJ is interrelated with gingival phenotype, which can vary according to age, gender, and ethnicity [24–27]. Considering that the average distance from CEJ to the alveolar crest is 1.5 mm (range: 1.08 to 1.71 mm) in stages I to III of passive eruption [8], the interproximal papilla height coronal to the interproximal CEJ is between 2–3.5 mm on average, and is greater in patients with a thick phenotype.

## 3. Altered Passive Eruption

Altered passive eruption (APE) is defined as a clinical entity that occurs in adults when a tooth has reached the occlusal plane and the gingival margin is located on the anatomic crown excessively occlusal to the CEJ in the absence of inflammation and gingival overgrowth [28,29]. "Altered" or "delayed" passive eruption is referred to as a morphological variant from a normal passive eruption process that exposes tooth structure secondary to apical migration of the gingival margin to a location at or slightly coronal to the CEJ by 20 years of age [11]. The passive eruption phase occurs in concert with the active phase of eruption (movement of tooth from its germinative position to its functional contact position with the opposing arch). Together, these two processes determine the position of the DGJ [10]. Clinically, APE is associated with excessive gingival display (gummy smile), short clinical crowns, and predisposition to gingival inflammation [30,31]. The prevalence of APE ranged from 12.1% to 29.5% and was found to be more common in patients with a thick periodontal phenotype [32,33]. However, there is a lack of clear diagnostic criteria to assess the magnitude of the amount of gingiva overlapping the anatomical crown leading to variations in the reported estimates of APE prevalence. The following are some of the criteria proposed to clinically define APE: 20% of overlapping of gingiva [34], 5–10 mm coronal extension of gingiva with respect to CEJ [35], approximation of the gingiva to the

maximum buccal convexity of a tooth [32], or >2 mm of a normal dentogingival dimension coronal to the CEJ [33].

Coslet et al. classified APE into two types (types 1 and 2) based on the width of existing keratinized gingiva in relation to the anatomic crown [36]. Type 1 is characterized by a noticeably wider gingival dimension from the gingival margin to the mucogingival junction (MGJ), and the mucogingival junction is usually located apical to the bone crest [13,37]. Conversely, in Type 2, the keratinized gingival width falls within the normal width and the mucogingival junction is usually at the level of CEJ. The normal width of keratinized gingiva was not clearly defined in the original paper by Coslet et al.; however, according to Bowers, Ainamo, and Löe, this generally accepted width is 1 mm of sulcus depth in addition to the width of attached gingiva of 3.0–4.2 mm in the maxilla and 2.5–2.6 mm in the mandible [13,37]. Coslet et al. further subclassified the two types based on the relationship between the alveolar bone crest and CEJ. In subgroup A, the distance from the alveolar bone crest to the CEJ will be 1.5 to 2 mm while in subgroup B, the bone crest level approximates the level of the CEJ. Recently, a proposal to rename subgroup B to "altered active eruption" was made, with the hypothesis that it is the failure of the active eruption phase leaving the CEJ positioned in close proximity to the alveolar bone crest [38].

Clinically, several commonly accepted criteria may suggest APE, including excessive gingival display (≥4 mm of gingiva-to-lip distance) [39], short clinical crowns (width/length ratio ≥0.85 in the absence of wear) [40], excessively flattened papilla base [34], and inability to detect the CEJ in a deep (>3 mm) gingival sulcus without concomitant pathological signs [11,41]. It is important to note that other conditions can contribute to excessive gingival display, including vertical maxillary excess, hypermobility of the upper lip, and anterior dentoalveolar extrusion [42,43]. Clinically, bone sounding or trans-sulcus probing is a commonly employed technique to confirm APE. However, it was suggested that bone sounding could be challenging in patients with APE, due to the long tightly attached epithelium and the proximity of the buccal bone crest to the CEJ (in subtype B) [44]. In summary, though several approaches exist, most experts agree that a facial negative GR of ≥3 mm is indicative of APE [11,30,33].

## 4. Gingival Enlargement

Any deviation from the normal dimensions of the gingival tissue signifies a pathologic event, and all changes involve some degree of inflammation induced by the bacterial biofilm [45]. "Gingival enlargement"—or interchangeably, "gingival overgrowth"—has many sources and can manifest with various clinical characteristics, including inflammatory enlargement of gingiva due to gingivitis, drug-induced gingiva overgrowth, gingival overgrowth associated with systemic conditions and diseases, gingival fibromatosis, and other neoplasm-associated gingival enlargements [45,46]. It could involve only the interdental papillae as a localized form or may involve marginal gingiva and papillae as a generalized clinical presentation. Dimensionally, it is difficult to quantify the severity and extent of gingival enlargement; however, indices have been developed to score the overgrowth severity [47–49]. Positive correlations were found between overgrowth severity and gingival inflammation, probing depths, calculus accumulation, plaque score, and the width of keratinized gingiva [50]. Gingiva enlargement is usually not associated with attachment loss, unless there is concurrent periodontitis; yet, it could be challenging to detect the CEJ in such cases to establish the true CAL. Inflammatory gingival enlargement (gingivitis) commonly originates as a slight ballooning of the interdental papilla and marginal gingiva, and continues to increase in size particularly when a local plaque retentive factor exists [51]. On the other hand, drug-associated gingival enlargement typically begins at the interdental papillae and is more frequently found on the labial surfaces of the anterior segment [52]. It frequently appears within 1 to 3 months following medication initiation, which makes the detection of early changes challenging. The degree of fibrosis and inflammation are associated with the dose, duration and type of drug, oral hygiene, and individual's susceptibility due to genetic and environmental influences [53]. Other pathologic gingival

overgrowth associated with systemic, hereditary, or neoplastic conditions are beyond the scope of this review.

## 5. Assessment of Clinical Attachment Loss

Clinical attachment loss is the distance from a fixed reference point, such as the CEJ, to the base of probable pocket. It is a clinical estimation of the loss of connective tissue attachment from the root surface [54]. The 2017 Word Workshop consensus recognized that a detectable interdental CAL at >1 site represents a "true" case of periodontitis [55]. However, the ability to detect early interproximal attachment loss may vary among operators based on their skill level and is influenced by the presence of local factors that impair the detection of CEJ [3,4]. Other factors may include anatomic variations [56], visual and tactile errors [57], presence of calculus [58], and restorative margins [59]. As a result, the diagnosis of incipient periodontitis based on CAL measured using a periodontal probe can be a formidable challenge for clinicians [60]. Secondly, the clinical pocket depth measurement, which is different from the histological pocket depth, is dictated by a variety of factors such as the type of the probe [61,62], probe tip diameter [63,64], probing force [65], probe angulation [66], contour of tooth surface [67,68], degree of inflammatory cell infiltration [4,69], and accompanying loss of collagen fibers [70]. In addition, the position of the gingiva margin, the reference point from which probing depths are made, is subject to extensive variability [57,71].

## 6. Clinical Assessment Approaches

### 6.1. Manual Instrumentation

CAL is determined by taking into account the probing depth and GR measurements. In the site where a negative GR exists, the distance from the gingival margin to the CEJ is subtracted from the probing depth to determine the CAL. Whereas at sites with positive GR (gingival margin located apical to the CEJ), CAL is determined by adding the PD with the positive GR value [2]. Locating the CEJ in the presence of negative GR can be particularly challenging, and there is a degree of guesswork involved in this indirect clinical approach that is subject to measurement errors and variations between examiners. CEJ location in such an approach is determined based on crown anatomy (crown length and ratio of length/width), curvature of the CEJ, and visibility of the adjacent CEJ. In the literature, measurement errors in the range of 30–50% and 20–40% have been reported for CAL and pocket depth measures using the indirect approach, respectively [72–75].

Alternatively, the clinician can directly "reflect" the papilla by the periodontal probe to visualize or "feel" for the CEJ to assess the amount of negative recession. However, in some scenarios, the CEJ may not be visible or may lack a clear demarcation, leading to visual or tactile errors [57]. On the proximal surfaces, the partial vertical course of the CEJ further increases the difficulty in locating the CEJ [76]. In addition, it is usually not feasible in daily clinical practice to complete a periodontal chart by direct approach due to its time-consuming nature. Vision-enhancement methods, e.g., the use of the operating microscope, equipped with high magnification (10–25×) and co-axial illumination, could assist in identifying the CEJ for better negative-recession determination, but the availability of this tool for clinical use is currently limited (Figure 2). Another "direct" way to detect true attachment loss is to use the periodontal probe tip running along the tooth crown in an apical direction until the bottom of pocket is reached. Attachment loss is measured by a mental note made of the extent of sliding movement withdrawn from the pocket bottom until the CEJ is detected [77]. The reproducibility of this approach is limited, mainly from the variation in individual tactile perception of CEJ. From a measurement-error point of view, direct measurements are less error-prone than the indirect approach. However, the intra- and inter-examiner variabilities are still very high and both these approaches can be time-consuming [78].

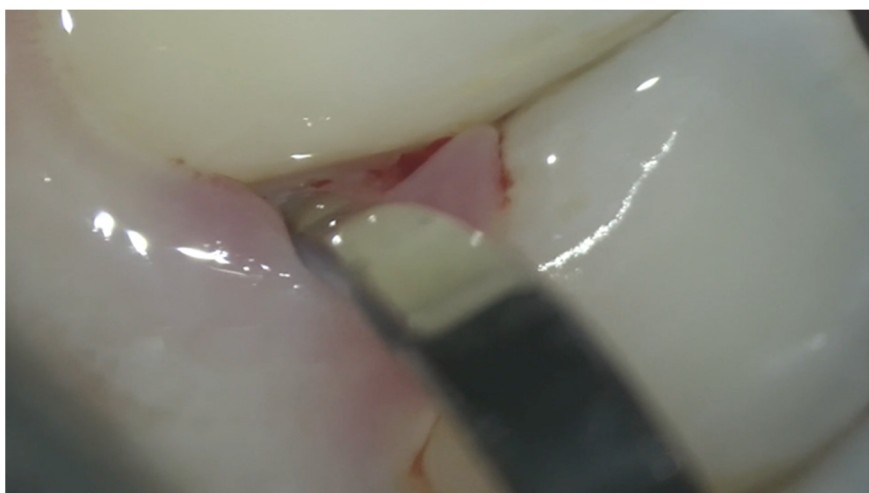

**Figure 2.** Visualization of negative recession using an operating microscope with magnification (10×) and co-axial illumination during ultrasonic scaling (Supplemental Video).

### 6.2. Automated Instrumentation

Several automatic electronic probes have been developed to address the high variability of PD and CAL. Jeffcoat et al. proposed a periodontal probe with automated CEJ detection function with controlled insertion force of 35 g [79]. A miniature accelerometer and position transducer processed the simultaneous measurements of the displacement and acceleration of the probe tip in order to identify the discontinuity in the slope that corresponds to the CEJ. Despite the high repeatability, it has been only used for research purposes [80]. Preshaw et al. modified an automated Florida probe and created a flange to detect CEJ [81], and it was proven to have increased inter-examiner consistency in detecting CAL [82,83]. It was reported that 2 to 3 mm cumulative change in CAL was required before a statistically significant loss of attachment level was detected by a manual probe [84], compared to a minimum detectable change of 1 mm for an electronic probe [85]. However, later evidence indicated that electronic probes do not offer a substantially advantage to reduce measurement errors [86–88]. Although the electronic probe can overcome errors and some of the limitations of manual probes [89], the manual probe is easier to use, less time-consuming, economical, and can be walked circumferentially to identify the deepest pocket. These factors have limited the widespread use of electronic probes [90,91].

### 6.3. Imaging Technologies

Radiographic techniques including intra-oral radiography and cone-beam computed tomography (CBCT) have been employed to assess the soft-tissue dimension in relation to the CEJ, but the scatter radiation and low-contrast resolution of soft tissue and CEJ outline limits their use in determining negative GR [92,93]. Alternatively, dental ultrasonography, a noninvasive and nonionizing modality, has been proposed to image periodontal tissues [94]. Ultrasonography functions by transmitting sound waves from the ultrasound transducer (probe) though a medium and then recording time-dependent reflections from tissues to determine its dimensions. Efforts have been made to develop an ultrasonic device to measure periodontal pocket depths, but the approach has failed to demonstrate reproducibility [95]. The device consisted of a thin probe inserted into the sulcus that directs the sound wave into the pockets in the presence of water (for coupling), and a computer algorithm then identifies the junction between the junctional epithelium and the connective tissue via the impedance difference of returning echo signals which infers the depth of the periodontal pocket. More promising outcomes in mapping the periodontal tissue dimensions have come with the evolution of transducers with higher frequency (higher image resolution). These have included studies in porcine models [96–99], human cadavers [100,101], and live humans [102–105].

In an early study, a dermatological ultrasonic transducer with a frequency of 20 MHz was shown to yield a better repeatability of linear measurements between an inscribed notch and the alveolar crest in pig jaws, when compared to the direct measurement and transgingival probing [96]. Chifor et al. also used a 20-MHz dermatological ultrasonic device to measure the periodontal ligament width, alveolar bone thickness, and gingiva thickness [99], and found significantly higher correlation among ultrasound, CBCT, and microscopy techniques [97]. Nguyen et al. used a linear broadband 8–40 MHz transducer that was designed for musculoskeletal and peripheral vascular exams to identify the distance from the gingiva margin to CEJ and alveolar crest. They observed a high degree of accuracy with the use of ultrasound when compared to CBCT measurements with less than 10% difference [98]. The same research team used a 20 MHz transducer to image the CEJ and found a high agreement of up to 97% between ultrasound and micro-CT images [106]; high inter-examiner reliability ($\geq 0.98$) in CEJ identification was also demonstrated [104].

Recently, dental ultrasonography has been modernized to a miniature-sized (comparable to a toothbrush) ultrasound transducer suitable for use in the oral cavity with higher-frequency (24 MHz) and high-resolution imaging (Figure 3). Its accuracy in the evaluation of periodontal and peri-implant tissue dimensions was validated by comparing these measures to direct bone sounding and CBCT [105,107,108]. Figure 4 provides an example of a cross-sectional scan at mid-facial and interproximal planes of a periodontal healthy maxillary central incisor using a modern commercially available ultrasound scanner (ZS3, Zonare, Mindray, Mahwah, NJ , USA) coupled with a 24 MHz miniature-sized imaging probe. In this example, the CEJ is clearly visualized and negative GR was measured to be 0.5 mm higher at distal papilla than at the mid-facial site. In contrast to probing, ultrasonography allows for noninvasive imaging of the CEJ and alveolar bone crest without the measurement variability caused by the inflamed gingiva tissues. We know from earlier studies that inflammation greatly impacts the probing depth measurement with the probe going past the junctional epithelium in inflamed sites while stopping coronal to apical termination of junctional epithelium in noninflamed sites [109–111]. Most of the time, clinical attachment levels by probing are within 1 mm of the true histological level of the connective tissue attachment [71]. Although current B-mode ultrasound imaging cannot directly determine the true CAL, it can aid in its calculation by accurately measuring the amount of negative GR. This can supplement the clinical probing depth to calculate CAL directly. Alternatively the distance from the CEJ-bone crest can be measured by ultrasound, then the CAL can be indirectly calculated by subtracting 1 mm of average dimension of connective tissue attachment [8] from the CEJ-bone crest distance. In addition, ultrasound-image modalities other than B-mode might be able to detect the junction between the detached and attached gingival tissues and are currently under active investigation.

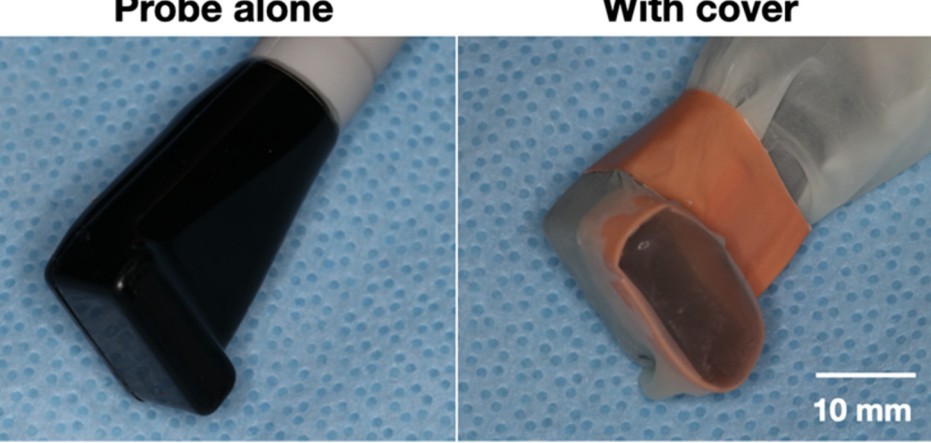

**Figure 3.** Modern miniature-sized ultrasound probe (transducer) with high frequency (24 MHz).

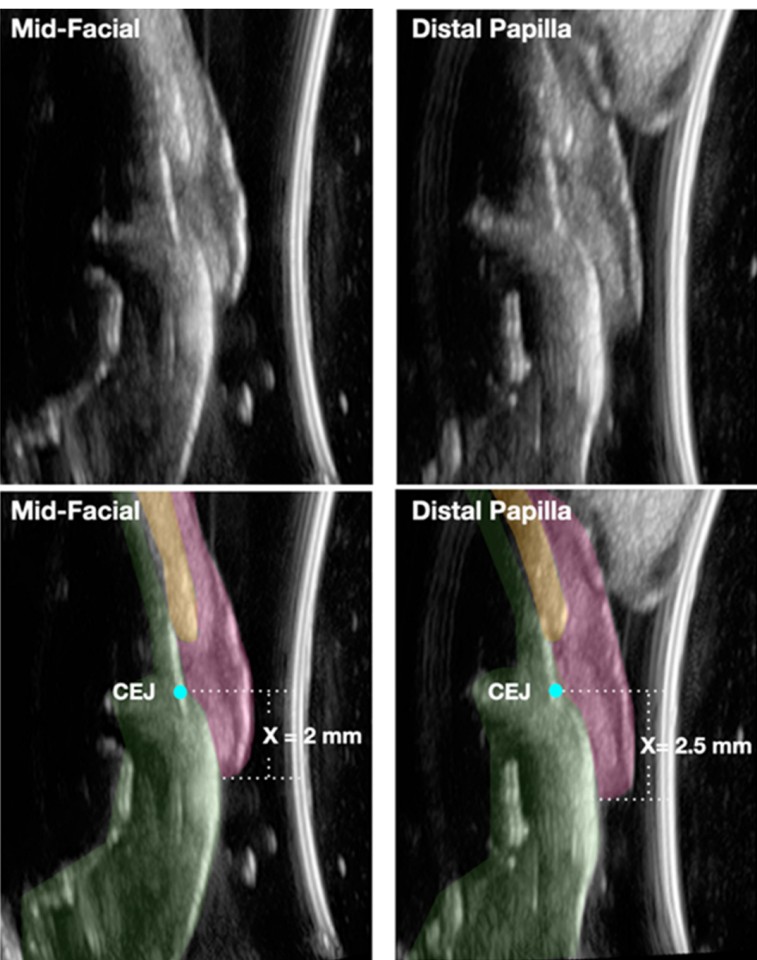

**Figure 4.** Example of an ultrasonographic cross-sectional image showing negative gingival recession on a maxillary central incisor. Colored areas represent different anatomic locations. Red: gingiva; yellow: alveolar bone; green: tooth. Cyan round point represents the CEJ. X measurements indicate the amount of negative gingiva recession.

Together, the advantages and disadvantages of different clinical approaches to assess negative GR are summarized in Table 2. Accurate assessment of the gingiva recession is a key component of periodontal diagnosis, especially in the incipient stage of the disease. Future research is warranted to differentiate the relationship between detached epithelium and connective tissue attachment using high-frequency ultrasound imaging.

**Table 2.** Summary of the methods to determine negative gingival recession.

|  | Advantages | Disadvantages |
|---|---|---|
| Periodontal probing | Easy, economical, quick | Prone to visual and tactile error |
| 2D radiography | Economical | Only detect interproximal CEJ, poor soft-tissue contrast resolution, errors related to image-acquisition angle |
| 3D radiography | CEJ is identifiable at bucco/lingual and interproximal sites | Ionization, expensive, poor soft-tissue contrast, lower image resolution |
| Ultrasound imaging | Noninvasive, nonionizing, chairside, providing digital image including dimension from gingiva margin to CEJ | Coupling agent needed, time-consuming, technique-sensitive |

### 7. Conclusions

- Measuring the amount of gingiva recession, both negative and positive, is an integral part of determining the clinical attachment loss.
- Detecting the CEJ has proven to be a challenge clinically when the gingival margin is coronal to the CEJ. To properly diagnose the amount of negative gingival recession, understanding the normal site-specific anatomy is the first step.
- With the aid of novel noninvasive and chairside ultrasound imaging and the high-magnification operating microscope, accurate and reproducible assessment of the negative gingival recession can become a reality that allows for early detection and intervention of periodontitis. These technologies could also prove to be valuable clinical and research tools in accurately detecting the amount of clinical attachment gain resulting from periodontal therapeutic modalities.

**Supplementary Materials:** The following supporting information can be downloaded at: https://www.mdpi.com/article/10.3390/app12147015/s1.

**Author Contributions:** Conceptualization, I.-C.W. and S.E.; writing—original draft preparation, I.-C.W.; writing—review and editing, I.-C.W., H.-L.C., G.K.J. and S.E. All authors have read and agreed to the published version of the manuscript.

**Funding:** This research received no external funding.

**Conflicts of Interest:** The authors declare no conflict of interest.

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
