# Peer review of "Assessment of Negative Gingival Recession: A Critical Component of Periodontal Diagnosis"

_applsci, doi:10.3390/app12147015_

Round 1
Reviewer 1 Report
Dear authors,
With a great pleasure I read an article that discusses in detail the normal anatomy of dentogingival junction and evaluation of clinical attachment loss methods. I think the article will be valuable to dentists of all fields. Though this is not a systematic review, provided material in this format is also valuable.
Reviewer 2 Report
The review is genuine, interesting and may presents an excellent evidence for clinical applications. However, the authors should address the following points:
* The authors should identify the type of review they're conducting (narrative, scoping...etc) in the abstract.
* The authors should add an introductory paragraph that outline the current problem, research question, existing research gap and objectives.
* Although Gargiulo study has established the bases of periodontal anatomy, several articles came afterwards to present new evidence and variable dimensions. Ethnicity, gender and age may play role in periodontal dimensions. Please add few sentences that covers that matters.
* The authors should add a section on clinical significance and another one on future direction of research and studies.
* The conclusion section should be summarized in bullets.
* The figures and tables should be inserted within text where they belong for proper flow.
* The authors may consider expanded discussion on each one of the four listed techniques.
Reviewer 3 Report
Dear Authors,
I have read and analyzed the manuscript “Assessment of Negative Gingival Recession: A Critical Component of Periodontal Diagnosis”.
The manuscript focuses on important issues regarding the periodontal diagnostic, the loss of periodontal attachment and the difficulties encountered in establishing a correct measurement. I consider it to be a well written manuscript, making clear, in a really honest way, the fact that calibration can represent a challenge in clinical assessment, an aspect which I consider that all practitioners should recognize and try to overcome.
I want to congratulate you for your work, it was a pleasure reading it; nevertheless, I have a few suggestions for your manuscript:
- An Abstract needs to be presented.
- Please, revise the typing along the manuscript. There are extra spaces and also lack of them between the reference brackets and the previous word.
- In my opinion, the Figures and the Tables would work better in the text body.
- Please, revise the Reference list for unity.
Sincerely yours,
